# Mycogenic Metal Nanoparticles for the Treatment of Mycobacterioses

**DOI:** 10.3390/antibiotics9090569

**Published:** 2020-09-02

**Authors:** Marta Filipa Simões, Cristiane Angélica Ottoni, André Antunes

**Affiliations:** 1State Key Laboratory of Lunar and Planetary Sciences, Macau University of Science and Technology, Avenida Wai Long, Taipa, Macau SAR, China; aglantunes@must.edu.mo; 2Bioscience Institute, São Paulo State University (UNESP), São Vicente, SP 11380-972, Brazil; cristiane.ottoni@unesp.br

**Keywords:** metal nanoparticles (MNP), silver nanoparticles (AgNP), filamentous fungi, mycogenic synthesis, tuberculosis (TB), *Mycobacterium tuberculosis* (MTb), nontuberculous mycobacteria (NTM)

## Abstract

Mycobacterial infections are a resurgent and increasingly relevant problem. Within these, tuberculosis (TB) is particularly worrying as it is one of the top ten causes of death in the world and is the infectious disease that causes the highest number of deaths. A further concern is the on-going emergence of antimicrobial resistance, which seriously limits treatment. The COVID-19 pandemic has worsened current circumstances and future infections will be more incident. It is urgent to plan, draw solutions, and act to mitigate these issues, namely by exploring new approaches. The aims of this review are to showcase the extensive research and application of silver nanoparticles (AgNPs) and other metal nanoparticles (MNPs) as antimicrobial agents. We highlight the advantages of mycogenic synthesis, and report on their underexplored potential as agents in the fight against all mycobacterioses (non-tuberculous mycobacterial infections as well as TB). We propose further exploration of this field.

## 1. Introduction

### 1.1. Mycobacterioses

Currently, there are almost 200 described species of the ubiquitous acid-fast bacteria of the genus *Mycobacterium* [1]. Mycobacteria can cause many different mycobacterioses, being the most worrying Tuberculosis (TB) [1], while common mycobacterioses can be caused by non-tuberculous mycobacteria (NTM). Some NTM such as *Mycobacterium abscessus, M. avium, M. kansasii, M. malmoense*, and *M. xenopi* can cause pulmonary diseases; others like *M. chelonae*, and *M. haemophilum* are able to cause disseminated diseases; furthermore, *M. fortuitum, M. marinum,* and *M. ulcerans* are able to cause skin, soft tissue, and bone diseases [2]. Even though there are some pathogenic species, NTM are opportunistic and considered as nontransmissible [3,4]. The distinction between pathogenic and non-pathogenic species is not always trivial as many of them share the same phenotypic and genotypic characteristics and have very limited differences [3,5].

Despite their similarities, NTM have lower human pathogenicity than mycobacteria from the *Mycobacterium tuberculosis* (MTb) complex [6,7] (species with 85–100% DNA homology with MTb, which include *M. africanum, M. bovis, M. caprae, M. canetti, M. pinnipedii, M. tuberculosis, M. microti,* or *M. mungi*, which are all pathogenic [3,7,8]). The fact that NTM share many infectious traits with the causing agents of tuberculosis (TB) allows them to be used in many research studies as model organisms of infection for this disease, with the advantage of being less pathogenic and faster growing species [7,9].

There is an increasingly large number of NTM species being isolated, identified, and described, yet many clinical isolates are lacking a species attribution [1]. This lack of assignment is an issue as treatments are specific for each species and can be extremely variable [6].

### 1.2. Tuberculosis

TB remains one of the top ten causes of death in the world. It is the infectious disease that causes the highest number of deaths, having a global distribution, and affecting all age groups and genders [10]. MTb, the causing agent of TB in humans, is particularly deadly due to some of its differentiating characteristics: (1) thick cell wall composed of peptidoglycan, arabinoglalactans, and mycolic acids; (2) slow growth rate; (3) ability to persist in non-replicating form for a long period; (4) capacity to cause latent-infection; (5) easy development of drug resistance; and (6) drug tolerance [11,12]. The main virulence factors responsible for these characteristics are: lipoarabinomannan (LAM), mannose-capped-LAM, lipomannan (LM), cord factor (Trehalose-6,6-dimycolate–TDM), phosphatidylinositol mannosides (PIMs), phthiocerol dimycocerosates (PDIM), phenolic glycolipids (PGLs), twin-arginine translocation (TAT) system, exported repetitive protein (Erp), and proteins from the ESAT-6 family [7].

TB can develop under two different forms: latent TB and active TB. Latent TB can last for very long periods (up to decades) and occurs when mycobacteria remain dormant. The World Health Organization (WHO) reports 1.7 billion people as being latently infected [10]. Latent TB infected people are asymptomatic and, if untreated, some of them progress to active TB at some point during their life [13]. Active TB occurs when mycobacteria actively multiply and spread [14].

According to the latest numbers, 10,000,000 people were diagnosed with TB in 2018; 484,000 were drug-resistant cases, with 377,500 being multidrug resistant (MDR) and 23,406 being extensively-drug resistant (XDR) [10,15,16]. Drug resistance is a major challenge to the diagnosis and therapy of TB. Furthermore, resistant TB leads to increased costs and hampered TB control and eradication [17,18].

MTb resistance was first reported around 1940–1950, shortly after the first use of antibiotic therapy against TB [19]. Contrary to what is often described for other bacteria, drug resistance in mycobacteria is not associated with horizontal acquisition of resistance genes. Instead, resistance in this group is due to the mutation of housekeeping genes coding for: (1) drug targets, (2) proteins involved in drug uptake, (3) efflux pumps, or (4) the activation of pro-drugs. The accumulation of those mutations, most commonly resulting from failed chemotherapy, seems to be the main cause of MDR in TB and other mycobacterioses [18].

#### 1.2.1. TB and Co-Infections

Mycobacteria are frequently reported as taking advantage of changes in host susceptibility to infection [3,4,5]. Furthermore, several infections increase the risk of developing TB, either due to the drugs to treat them or the weakening of the immune system, with some of them becoming co-infections. 

As an example, the human immunodeficiency virus (HIV) is commonly associated with TB and other mycobacterioses, with mycobacteria becoming opportunistic infectious diseases in HIV-positive patients [10]. Such co-infected patients have an increased risk of mortality. Furthermore, disseminated TB is associated with HIV-induced immunosuppression (*i.e.*, in patients undergoing antiretroviral therapy and with low CD4 cell counts) [6]. Even though TB is usually a slow-progressing disease, disseminated TB progresses fast, making the right and effective therapy an urgency [20]. The co-infection of TB and Acquired Immune Deficiency Syndrome (AIDS) turns both infections into more complex challenges without easy solutions.

Other infections, such as those caused by some fungi (namely *Candida* spp. [21], *Aspergillus* spp. [22]), are also opportunistic, and take advantage of the weakened immune system of TB patients. The long and complex treatment for TB, together with the infection development, makes TB patients particularly susceptible to fungal infections. Aspergillosis, for example, has become a common co-infection among patients with pulmonary TB [22]. These two infections have an unfortunate close relation with similar clinical symptoms. Therefore, aspergillosis has long been reported to lead to TB misdiagnosis [23,24,25].

When there are co-infections or direct links between TB and inflammation of cells due to other diseases, further challenges become evident: therapeutic limitations, acquired resistance, toxic side-effects, and drug-drug interactions [26,27]. A disease for which TB has been referred as a risk factor is lung cancer, since it can cause alterations in the lungs, which might become a cause for later malignant cell changes [27].

#### 1.2.2. TB and COVID-19

The coronavirus disease 2019 (COVID-19) pandemic is having a huge social impact. Outcomes of this pandemic include the reduction or suppression of certain healthcare infrastructures and their access, mainly due to lockdown and other control measures (quarantine of suspected cases, isolation of infected patients, and contact tracing). As a result, less cases of TB will be detected, and its infection rate will increase [28]. The lockdown also affects the production and transport of drugs and supplies, and limits the access to healthcare services, causing the disruption of treatments of certain diseases, which will be particularly negative for people with drug-resistant TB [29]. This will certainly worsen the problem of resistance in TB and other infections. Confinement also facilitates the contact of infected and non-infected members that share the same house, thus increasing household transmission of TB [30].

Both TB and COVID-19 present similar and non-specific clinical features—fever, cough, and dyspnea, or breathlessness [15,31,32]. When COVID-19 testing is not available, their similarities can easily lead to misdiagnosis and ineffective treatments [30,32,33,34]. Another potential challenge is the stigmatization of TB patients (for coughing), due to the fear of COVID-19 [30,31]. These patients become afraid of visiting healthcare services, and many (infected with TB) end-up not being properly diagnosed within a timeline that would contribute to control the infection [31].

Cases of patients simultaneously infected with TB and COVID-19 present other risks: when radiology data is not available, TB might not be diagnosed; COVID-19 therapy can reactivate latent TB; pre-existing TB, especially if active, will worsen the clinical state of COVID infection; and the simultaneous therapy for both infections can lead to drug-drug interactions and added hepatotoxicity [32].

The WHO has recognized the impact of COVID-19 on TB and has issued a note on how to tackle this serious issue [15]. Nevertheless, some of these negative consequences will be unavoidable and long-lasting [16,35]. As a result of the pandemic, the next five years are expected to reverse the trend of the past decades [29] and will lead to an increase in TB incidence and mortality [36].

### 1.3. Current Solutions

The common treatment for TB is a multidrug combination of first-line drugs, consisting of two months of rifampicin (RIF), isoniazid (INH), pyrazinamide (PZA) and ethambutol (EMB), followed by four months of RIF and INH [6,10,37]. These standard combinations are reasonably low-priced and effective against sensitive mycobacterial strains [9]. However, due to the long duration and toxic side-effects of the therapy (especially when treating drug-resistant cases), many patients end-up not following the complete treatment, thus increasing the chance of reemergence of the disease and development of resistance [10,14,38].

Unfortunately, the six-month treatment is not effective against MDR nor XDR TB [39]. MDR TB implies simultaneous resistance to RIF and INH, whereas XDR TB implies an additional resistance to a quinolone and at least one injectable drug [17]. The increasing number of cases of MDR and XDR TB lead to the use of second-line drugs which are more toxic and less effective than first-line ones [17]. The treatment of MDR TB is more expensive, longer (can go up to 24 months), and uses a combination of at least five drugs with many harmful secondary effects (such as hepatotoxicity) [40]. Failure of treatment hardens the challenge of fighting TB because it increases the infection rate and raises mortality [39]; 70% of XDR TB patients have been estimated to die within a month of diagnosis [14]. Therefore, all options that can contribute to improve efficiency of antimycobacterial activity and reduce toxicity should be fully explored.

Carrier or delivery systems, such as liposomes and microspheres, have been developed for the sustained delivery of anti-TB drugs and have shown better chemotherapeutic efficacy [17,41]. Conjugating existing drugs with NPs is another strategy that has great potential for the treatment of MDR TB. This is the case of mesoporous silicon NPs conjugated with ethionamide (ETH), a second-line drug, that has increased activity against MTb when compared to ETH alone [42]. Hakkimane and colleagues synthesized NPs with poly lactic-co-glycolic acid polymers encapsulating RIF or INH, and found both formulations to be more effective than RIF or INH against MTb, having a higher activity and requiring a lower drug concentration [39].

Chemical synthesis of new drugs, such as prodrugs derived from first-line agents, is another strategy also studied as an alternative solution to fight TB where different chemical structures might circumvent previous resistance [43]. Some peptides, part of the host first-line of defense and produced by the innate immune response, have been identified as having antimicrobial activity, presenting good biocompatibility and low probability of leading to microbial resistance [44]. These have been described as having a direct action against microorganisms, by creating cytotoxic pores on their cell walls, and an indirect one, by modulating the host immune system through upregulated secretion of pro-inflammatory cytokines and chemokines and contributing to contain infection [44,45,46]. Therapies using these antimicrobial peptides have been investigated, and synthetic peptides have been shown to have great potential and activity against pathogens, combined with decreased toxicity [46,47].

However, despite all this research and promising leads, their application to TB therapy remains underwhelming and a completely effective TB vaccine has also not yet been developed [14].

## 2. Methods

The current research was developed according to the PRISMA Protocol. A literature search was performed on three databases (Research Gate (www.researchgate.net), Google Scholar (https://scholar.google.com), Academia (www.academia.edu), and further cross-checked and complemented with searches on Web of Science (www.webofknowledge.com), Scopus (www.scopus.com/home.uri), and PubMed (https://pubmed.ncbi.nlm.nih.gov). The search criteria were based on the following key terms: nanoparticle, mycogenic, biologic, mycobacterium, and AgNPs. Searches with all possible combinations of the referred terms were performed. Selected articles included all of those mentioning the use of mycogenic metallic nanoparticles against mycobacteria, as well as the most recent (≈last decade) manuscripts on AgNPs. The last date for these searches was 4 August 2020. Any article matching the searching criteria was checked and used in our review.

## 3. Results

The rise of microbial resistance against antimicrobial drugs has encouraged and promoted nanotechnology research as a potential viable source of solutions [48]. NP-based systems are able to circumvent many of the challenges related to mycobacterial infections since they can target the infected cells and act directly on the cell wall of intracellular pathogens, as is the case of MTb when it infects macrophages in pulmonary TB [26].

It is common knowledge that metal nanoparticles (MNPs) have antimicrobial activity against a multitude of microorganisms [40]. MNPs represent promising potential solutions against many infections and resistance to traditional drugs. They use different mechanisms of action from those identified for common drugs, exhibit activity against many microbial resistant species and strains, and target several biomolecules interfering in the development of microbial resistance [49]. The mechanisms that have been reported to explain the effects of MNPs on microbial cells include: DNA damage, protein damage, mitochondrial damage, attachment to 30S ribosome subunit, oxidation of cellular components, release of metal ions, damage to the proton efflux pump, disruption, or prevention of biofilm formation, disruption of cell membrane, disruption of transmembranar electron transport, and production of reactive oxygen species (ROS) [17,49]. Some of these mechanisms vary depending on the target species and the specific characteristics of the NPs [17].

### 3.1. Silver Nanoparticles (AgNPs)

Silver (Ag) has the highest reflectivity of all metals [50]. It has been used for centuries as an antimicrobial agent [40]. Throughout history, civilizations incorporated silver into daily life objects to avoid spreading diseases, for example, in ancient times, silver containers were used to keep water potable and prevent wine spoilage. However, once antibiotics were discovered, the use of silver for its antimicrobial activity decreased [50]. Nowadays, nanotechnology is responsible for a resurgence in the exploration of silver for these purposes. 

AgNPs are the most widely used MNPs due to their potential as therapeutic agents and antimicrobial agents, showing activity against almost 700 pathogens [19,51]. AgNPs are used in a vast number of different products and applications (namely textiles, cosmetics, food packaging, medical appliances, pharmaceutical ointments among many others) [40,50,52]. They have a significant impact on respiratory medicine and can be applied against a broad range of microbial infections [27,52]. AgNPs are an alternative way to overcome drug resistance [53], mostly due to their particular characteristics: small size, even morphology, and capacity to interact with biomolecules [19]. Furthermore, AgNPs have been reported as having good conductivity, chemical stability, catalytic activity, cytotoxic effect on cancer cells, and antimicrobial activity [49,54].


**AgNPs against mycobacteria**


AgNPs have proven antimycobacterial activity, but this activity is highly variable, dependent on several parameters [52], and also on target species [26,53,55]. Smaller AgNPs tend to have higher activity due to their larger surface/area, which allows them to release higher amounts of silver ions and inhibit microbial growth [26,56]. In addition, higher concentrations lead to higher activity [55,57]. Some studies also point to the potential relevance of AgNPs shape, although this has only been rarely analyzed. Triangular-shaped AgNPs seem to be more effective against *Escherichia coli*, likely due to an increase in positive charges and more active facets induced by this morphology [58,59]. Although AgNPs’ antimicrobial mechanism of action is not fully clear, we know that they kill by contact and ion release [19]. It was also detected that the antimycobacterial activity of AgNPs is higher than other metallic NPs, making them a favorite focus on this research field [40,56,60].

Furthermore, in addition to antimycobacterial properties *in vitro*—when applied directly on mycobacteria, AgNPs have also shown *ex vivo* activity by suppressing innate responses of infected macrophages, induced by mycobacteria [40]. Mycogenic AgNPs have also been reported to have anti-inflammatory activity [54]. All of the characteristics of AgNPs make them an undoubtedly easy focus of nanotechnology research.

An overview of recent research on the use of AgNPs against mycobacterial species, not including MTb, is presented below (Table 1). This overview excludes MTb which will be analyzed afterwards due to its epidemiological relevance.

We can note that the majority of these studies relied on chemically synthesized AgNPs. Furthermore, they have focused on a reduced number of species, the most common being *M. smegmatis* and *M. bovis*. The reasoning for this focus is the fact that *M. smegmatis* is the most used model organism for the study of mycobacteriosis and TB, as they are safe and non-pathogenic mycobacteria as well as fast-growing and easy to manipulate genetically [61]. *M. bovis* (strain BCG) is a slow-growing mycobacterium, whose choice as a model is mostly due to its placement within the MTb complex, being an attenuated strain and a biohazard level 2 microorganism (*versus* biohazard level 3 for MTb) [62].

Only a few studies have looked specifically into the production of AgNPs against MTb (Table 2). These include AgNPs produced via physical-chemical methods, as well as biological ones (mostly using parts of plants), and consist of a wide-range of NP sizes and tested strains (avirulent, virulent, isolated from clinical samples, drug-sensitive and drug-resistant—MDR and XDR).

The research made so far on the use of AgNPs against MTb has proved that these are effective. The lack of standardization among all research studies makes it more difficult for us to compare them across, although they provide some relevant insights into the use of this type of NPs against MTb. The general consensus that smaller sized AgNPs are more active against bacteria is also valid for mycobacteria [56,71]. This is the case, for example, of the smaller BSA-AgNPs (Table 2) that showed higher activity than the produced PVP-AgNPs [64].

Furthermore, these studies present promising results with cases of activity reported against MDR and XDR strains [19,26,71,72,78], as well as activity against intracellular MTb [56,73]. This is especially relevant for latent TB where MTb remain inside granulomas [79], where they adapt and thrive under adverse conditions, such as nutrient deprivation and hypoxia [3].

### 3.2. Mycogenic MNPs

Mycogenic processes are biological processes developed by fungi, mostly filamentous fungi (molds)—fungi that form mycelia. These organisms have the capacity to accumulate metals by sequential action of reductase enzymes (such as NADPH-dependent nitrate reductase), leading to the reduction of metal salts and final production of metal nanoparticles (MNPs) [58]. As such, they can be affected by several parameters (Table 3).

These parameters can be adjusted and optimized. They are variable for some processes and for the species used for the production of MNPs [52]. Interestingly, the MNPs used in *in vitro* assays don’t have their antimicrobial activity affected by temperature or pH [17]. However, the full extent of the effects of these parameters (Table 3) is not yet completely understood and requires further research.

Even though mycogenic MNPs can be produced intra- or extracellularly, the latter production process is more appealing. Extracellular production involves fewer steps and does not require cell disruption to release the MNPs or complex washing steps to recover and purify them [17]. Many studies have reported and listed the capacity of different fungal species to synthesize MNPs, from ubiquitous groups, to extremophiles, and ranging from yeasts, to filamentous fungi and mushrooms [18,54,58,80,81,82,83,84,85,86]. Despite this wide diversity, most mycogenic processes rely on filamentous fungi and follow a four-main steps method (Figure 1).

Generally, the process includes the growth of fungal biomass (steps I and II in Figure 1), followed by a sequential shorter incubation of that biomass in water (step III in Figure 1), and then the mixture of this supernatant with a chemical precursor (step IV in Figure 1). The fungal metabolites act as reducing and stabilizing agents, and lead to the reduction of metal ions and agglomeration of metal atoms, which result in MNPs. The MNP synthesis is noticeable by the change in the color of the suspension and can be further confirmed by UV-visible spectroscopy, which detects changes in the optical properties of the MNPs, by reading the absorbance of the surface Plasmon resonance bands (usually localized at a wavelength of 400–450 nm). The MNP suspension should then be cleaned and purified either by filtration, dialysis, or ultracentrifugation.

The last few years have shown a considerable amount of research and development in the use of mycosynthesis of the most studied MNPs—AgNPs (Appendix A). These further highlight their wide applicability and support the use of mycogenic AgNPs as antimicrobial agents as well as for all other applications of general MNPs [48,55,87,88,89,90,91,92,93,94,95,96,97,98,99,100,101,102,103,104,105,106].

Most mycogenic AgNPs studied are spherical although some variations, mostly due to the formation of clusters, can also be found [41]. In most mycogenic processes (those following the method shown in Figure 1), the common concentration of the precursor—silver nitrate (AgNO_3_)—is 1 mM [48,89,103,106]. Within the size variation described, mycosynthesized AgNPs when produced with enzymatic stimulation (from co-cultures) form smaller and more active NPs [106].


**Mycogenic AgNPs against mycobacteria**


Mycogenic synthesis of AgNPs combines the best of both worlds: the most effective MNPs that nanotechnology has developed together with the green production process mediated by fungi [80]. These mycogenic processes allow us to obtain smaller sized NPs than those produced as a result of chemical or physical synthesis, which in turn allow a better control over production size, being less size-variable [83]. Furthermore, synthesis parameters can be altered in order to change their size and therefore activity [52,54]. So far, little has been explored combining the use of mycogenic AgNPs and the fight against TB and other mycobacterioses, but the efforts presented in Table 4 show the potential to exert some control over this infection and its increasing resistance.

In general, mycogenic MNPs are mostly monodispersed, with well-defined size and shape (which is most commonly spherical) [85]. Moreover, the mycogenic AgNPs reported to be successful against mycobacteria were obtained from a diverse range of fungi, which included yeasts and species within different families of filamentous fungi (Table 4). This supports the assumption that fungi constitute a potential source of MNPs relevant for fighting TB and other mycobacterioses.

Biological AgNPs are considered relatively safe and less toxic (less cyto/genotoxic *in vivo*) than chemically synthesized AgNPs [110,111]. Their toxicity depends on the concentration used [112]. In addition, generally, all AgNPs show increased toxicity once dissolved and after losing their spherical structure [49].

### 3.3. MNPs (Other Than AgNPs) against Mycobacteria

Transition metals (the 38 elements in groups 3 through 12 of the periodic table) are recognized as the most suitable elements for the synthesis of MNPs [49]. Within these, NPs incorporating silver are the most widely studied and used. The last few years have seen an increase in testing some of the other transition metals and others as a way of uncovering new MNPs and exploring their potential use as antimicrobial compounds (Table 5).

The study of MNPs with alternative metals against mycobacteria is arguably less dynamic. Nonetheless, a few researchers have investigated the use of elements such as copper, gallium, selenium, titanium, zinc, or even bimetallic alternatives. Within these, copper and zinc seem to be the two transition metals most commonly explored, with gallium (a post-transition metal) also being frequently studied.

Exploring these alternatives can be relevant to better understand which MNPs have lower probability of developing toxicity.

### 3.4. Synergistic and Complementing Effects of MNPs against Mycobacteria

Many studies have focused on the use of MNPs complemented with other substances or different MNPs (Table 6). Such combined uses of MNPs have often proved to be synergetic, increasing the antimicrobial activity of its individual components.

Combination therapy for mycobacterial infections can increase the potential activity of MNPs, contribute to decrease the effective dose of antibiotics potentiating them; reduce side effects, drug toxicity, and MNPs toxicity; enhance bioavailability; and enhance solubility and retention time [103,124].

The exact interaction mechanisms aren’t always fully understood. Generally, negatively charged microbial surfaces attract positively charged MNPs due to electrostatic interactions. The MNPs then establish bonds with the cellular membranes, disrupting the cell walls and making them more permeable. As a consequence, microorganisms become more sensitive to drugs [49]. The increased activity from the conjugation of MNPs with antimicrobial peptides, for example, might be due to the higher membrane permeability of the peptides, which then help deliver MNPs into the microbial cells [44]. Another example of increased antimycobacterial activity is the combination of AgNPs with chloroform, which is due to chloroform’s ability to remove lipids and rupture the mycobacterial cell wall [41].

One should also note that the conjunction of biomolecules such as peptides or chitosan results in increased antimycobacterial activity, but this effect is limited after mycobacteria are internalized by macrophages [7]. More effective strategies rely on combining NPs with classical anti-TB therapeutics that ensure both extra- and intracellular activity, although only a few *in vivo* studies have explored this option [7].

## 4. Discussion

### 4.1. Advantages of Using Mycogenic MNPs

The increased applications of MNPs in the medical field demand more biocompatible, safe, and effective nanostructures with less hazardous byproducts of synthesis reactions [83]. Mycogenic MNPs (as well other biological MNPs) are regarded as safe, less toxic, biocompatible, eco-friendly, and cheaper alternatives, with lower consumption of energy and higher yields when compared with physical-chemical synthesis [56,106].

Fungi are more efficient than most microorganisms when it comes to the biological production of MNPs [51]. This is due to the fungal capacity of producing a high number of bioactive metabolites, accumulating metals and having enhanced processes [54,125]. As additional advantages, fungi are easy to manipulate, easy to grow, do not require complex nutrients, have high production of biomass and metabolites, and have high wall-binding capability and high metal uptake [52,80,83,85,89,106,126]. Many of the fungal metabolites involved in the mycogenic synthesis also cap the MNPs, conferring a higher control of size and stability. In addition, because most mycogenic processes are extracellular, there is no requirement for additional steps, or downstream processing, to release the MNPs for further processing or use [49,80]. Moreover, extracellular mycogenic processes facilitate handling and scale-up, and mycelia from filamentous fungi are more resistant to agitation and pressure making them more suitable for large-scale synthesis in bioreactors and chambers [54,106].

Looking specifically at mycogenic AgNPs, a recent study observed that they were more active against pathogenic bacteria than chemically synthesized AgNPs [55], further highlighting the advantages of their use. A wide variety of fungal genera are recognized as being able to precipitate AgNPs (namely *Aspergillus*, *Fusarium*, *Penicillium* and *Verticillium*) [126]. However, there is a much wider fungal diversity that remains completely unexplored. Current estimates point to the existence of 1.5–3.8 million fungal species on Earth with only 120,000 (3–8% of the total) being validly described, leaving much to be discovered, isolated, and characterized [127,128]. The fungal kingdom is therefore underexplored with only a small percentage of total species already surveyed regarding their capacity to mycosynthesize MNPs. Produced MNPs can have a very diverse range of characteristics and biological activities which are dependent on the formation process and the enzymatic profile of each fungal species [90].

### 4.2. Future Focus of Research

Despite the well-recognized advantages of MNPs as one of the best alternatives against antimicrobial-resistant strains of mycobacteria and other taxa, several challenges and opportunities are ahead of us. Given that the effects of NPs result from a combination of multiple, synergistic mechanisms, the potential development of resistance against them is more arduous and less likely [124]. One should note, however, that NPs are unlikely to offer a full, definitive solution, and their misuse should be avoided, as it can lead to further issues. A recent report pointed to a case of an AgNPs and AgNO_3_ resistant mutant strain of *M. smegmatis*, developed after one single exposure and associated with increased MIC for INH [129]. This unexpected case seems to be a rare event but further highlights our need to study and understand MNPs. The focus of such future studies should be on testing new strains, discovering new NPs, and clarifying their synthesis and mode of action as antimicrobial agents.

Lines for potential novel discoveries on mycogenic MNPs include the study of endophytic and extremophilic strains. While the former has increased its relevance within the last years (Table 2), the latter is still restricted to a very small number of studies and is focused on other applications rather than their use as antimicrobial agents [130].

The application of new approaches such as synthesis optimization via statistical methods (such as central composite design and response surface methodology) has been proposed in a few studies and showed some promise but remains mostly unexplored [98,101]. Such approaches are expected to significantly reduce the number of different lab tests required and lead to a quicker optimization of MNP production, thus warranting a closer look.

Like with any other drug, microorganisms might develop resistance to MNPs [40,131], so it is essential to thoroughly investigate all aspects related to their application. This will allow us to understand and manipulate all caveats regarding their synthesis and antimicrobial mode of action, and standardize synthesis methodologies to attain best scale-up yields. Furthermore, as mentioned by Tăbăran et al. [7] in relation to AgNPs, there are a few other therapeutic obstacles to overcome: “poor delivery, variable intramacrophagic antimycobacterial efficiency, and residual toxicity”. The same applies to all other MNPs effective against mycobacteria.

Investing in exploring the capacity of new fungal species to mycosynthesize MNPs is a potential source of therapeutic alternatives. This requires the investment in isolation and bioprospection of uncommon and unexplored or under-explored environments. The great resource of new fungal species to be discovered might bring to light species with even more effective and better capabilities. There is much work to be done and much to be explored, but the prospects of mycogenic MNPs are very promising.

## 5. Conclusions

Mycobacterioses are an increasing public health threat and it is imperative to develop new and better solutions taking into consideration all options to control them. The scenario is particularly bleak for TB. Innumerous actions and campaigns over the recent decades have contributed to a tendency to control the infection, but TB is still neglected by many sectors. Worrying recent estimates [132]) predict that COVID-19 containment measures will exact a heavy toll on health services and therefore an increase on infections such as mycobacteriosis and TB cases.

MNPs constitute a viable option to aid us to face this threat. This is supported by the many studies done on the different MNPs synthesis processes and their vast array of applications. Furthermore, mycogenic processes of MNPs production, namely those relying on the use of filamentous fungi, are particularly promising. Among its many advantages, the use of this production method is simple, quick, low-cost, and eco-friendly. This nanotechnology could help in killing mycobacteria, lowering drug doses and therapy periods, which in turn would help control infections.

Despite their benefits, mycogenic MNPs remain an underexplored weapon to fight TB and other mycobacterial infections. Furthermore, mycobacteria and MTb in particular have a complex pathogenesis that is not yet fully understood. Considering that, according to Sarkar et al. [73], exposure to MNPs can suppress immune responses induced by mycobacteria, more efforts are necessary to understand and successfully transfer this nanotechnology to the patients with guaranteed safety, no toxicity, biocompatibility, and effectiveness.

## Figures and Tables

**Figure 1 antibiotics-09-00569-f001:**
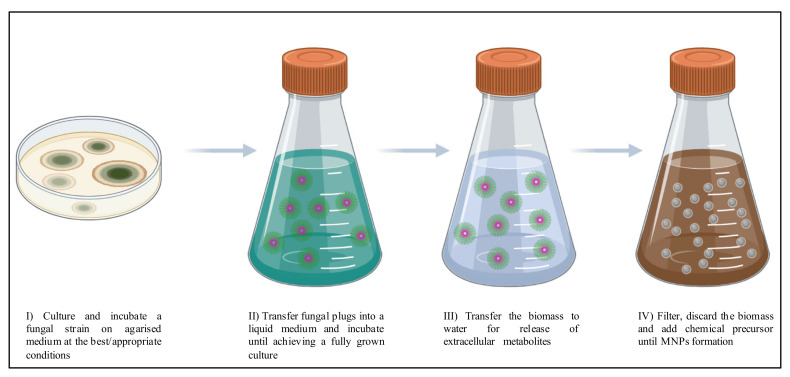
Main steps in mycogenic synthesis of MNPs.

**Table 1 antibiotics-09-00569-t001:** AgNPs with reported activity against mycobacteria—excluding *Mycobacterium tuberculosis*.

AgNPs Size	AgNPs Shape	Synthesis Process	Highlights	Ref.
20–25 nm	Spherical	Chemically synthesized	Activity against *M. smegmatis* (strain mc^2^155, MIC = 0.46 μg/mL) and *M. bovis* (strain BCG, MIC = 1.1 μg/mL)	[63]
5–45 nm	Spherical	Capped with bovine serum albumin (BSA), and poly-n-vinyl-pyrrolidone (PVP)	Activity of BSA-AgNPs and PVP-AgNPs against *M. xenopi*	[64]
55 and 278 nm	Spherical	Stabilized with chitosan	*In vitro* activity against *M. smegmatis* mc^2^155 and antibiofilm activity, *ex vivo* (raw 264.7 macrophages) antimycobacterial activity, noncytotoxic to macrophages, disruption of mycobacteria cell wall	[65]
25–45 nm	Spherical and monodispersed	Chemically synthesized with photo-irradiation	Activity against *M. bovis* (strain BCG)	[66]
≈20 nm	Spherical and monodispersed	Synthesized with 1% starch	*In vitro* antimycobacterial activity against *M. smegmatis* mc^2^155, *ex vivo* (raw 264.7 macrophages) antimycobacterial activity, noncytotoxic to macrophages, inhibition of biofilm formation	[67]
≈70 nm	Spherical with agglomeration	Chemically synthesized	Bactericidal effect against *M. smegmatis* when the AgNPs were complemented with 2% chloroform	[41]
6.9–18.3 nm	Spherical	Commercially acquired	Bactericidal effect when applied as coatings on polycarbonate membranes, against biofilms of *M. smegmatis*, *M. avium*, and *M. marinum*	[53]
30–130 nm	Spherical-oval	Biologically synthesized from leafs of *Ipomea carnea*	Activity against *M. smegmatis*	[68]
NR	NR	Biologically synthesized from leaves of *Psidium guajava* L.	Activity against *M. smegmatis* and *M. pheli*	[57]
1–5 nm	Spherical-oval	Chemically synthesized	Activity against *M. bovis* (strain BCG)	[56]
8–12 nm	Spherical	Biologically synthesized from *Acinetobacter* sp.	Activity against *M. bovis* (strain BCG)	[56]
<50 nm	Spherical	Chemically synthesized	Activity against *M. avium* subsp. *paratuberculosis*	[69]
38–52 nm	Spherical	Biologically synthesized from flowers of *Catharanthus roseus* L (apocyanaceae)	Activity against *M. smegmatis*, with 71% of growth inhibition	[70]
≈50 nm	Tetrahedral	Chemically synthesized	Activity against *M. bovis* (strain BCG, MIC = 4 µg/mL) and clinical isolates of *M. bovis* (MIC = 4–32 µg/mL)	[26]

BCG, Bacillus Calmette-Guérin; MIC, minimum inhibitory concentration; NPs, nanoparticles; NR, not reported; Ref., reference.

**Table 2 antibiotics-09-00569-t002:** AgNPs with reported activity against *Mycobacterium tuberculosis*.

AgNPs Size	AgNPs Shape	Synthesis Process	Highlights	Ref.
5–9 nm	Spherical	Conjugated with bovine serum albumin (BSA)	Activity against several drug-sensitive MTb (strain H37Rv) and clinical isolates of MTb	[64]
6–45 nm	Spherical	Capped with poly-n-vinyl-pyrrolidone (PVP)	Activity against several drug-sensitive MTb (strain H37Rv) and clinical isolates of MTb	[64]
10–20 nm	Spherical	Biological synthesis from extracts of cucumber (*Cucumis sativus*).	Activity against several drug-sensitive MTb (MIC = 7.8–12.5 µg/mL), MDR strains (MIC = 12.5 µg/mL), a XDR strain (MIC = 15.6 µg/mL), and NTM strains (MIC = 25 µg/mL)	[71]
NR	NR	Biologically synthesized from leaves of *Psidium guajava* L.	Activity against an avirulent MTb	[57]
≈70 nm	Spherical with agglomeration	Chemically synthesized	Bactericidal effect against MTb, only observed when complemented with 2% chloroform	[41]
>200 nm	Spherical and cubic	Biological synthesis from aqueous extracts of fresh *Moringa oleifera* leaves and *Allium cepa* bulbs	Activity against MTb strains (H37Ra, a wild type drug-sensitive, and a MDR strain)	[72]
1–5 nm	Spherical-oval	Chemically synthesized	*In vitro* activity against active MTb (strain H37Ra, MIC = 1.31 µg/mL) and dormant (MIC = 2.04 µg/mL), as well as *ex vivo* (in THP-1 cells) active MTb (MIC = 1.97 µg/mL) and dormant MTb (MIC = 2.18 µg/mL)	[56]
8–12 nm	Spherical	Biologically synthesized from *Acinetobacter* sp.	*I**n vitro* activity against active and dormant MTb (strain H37Ra), as well as *ex vivo* (in THP-1 cells) MTb (MIC > 2.56 µg/mL for all conditions tested)	[56]
20–110 nm	Spherical	Commercial AgNPs with surface modifications (citrate or poly-n-vinyl-pyrrolidone (PVP) coated)	*Ex vivo* (in human monocyte-derived macrophages) activity against MTb (strain H37Ra) Suppression of MTb-induced expression of IL-1β, IL-10, and TNF-α mRNA	[73]
38–52 nm	Spherical	Biologically synthesized from flowers of *Catharanthus roseus* L (apocyanaceae)	Activity against MTb, with 57% of growth inhibition	[70]
50–200 nm	Spherical and polygonal	Biologically synthesized from fruits of *Coriandrum* *sativum*	Activity against MTb (strain H37Rv, MIC = 1.56 μg/mL)	[74]
15–45 nm	Spherical and with face centerd cubic geometry	Biologically synthesized from leaf extract of *Plumbago auriculata*	*In vitro* activity against MTb (MIC = 1.6 μg/mL)	[75]
20–56 nm	Spherical	Biologically synthesized from flower extracts of *Sesbania grandiflora*	*In vitro* activity against MTb (strain H37Rv, MIC = 12 µg/mL)	[76]
32–50 nm	Quasi-spherical	Biologically synthesized from leaves of *Leucaena leucocephala* L.	Monodispersed with activity against obtained for MTb (strain H37Rv, MIC = 125 μg/mL).	[77]
10–70 nm	Spherical and polyhedral	Biologically synthesized from *Pseudomonas hibiscicola*	Polydispersed, with activity against MTb (strain H37Rv) and a clinical MDR MTb strain (MIC = 1.25 mg/mL for both strains)	[78]
≈50 nm	Tetrahedral	Chemically synthesized	Activity against MTb (MIC = 1 µg/mL), clinical isolates of MTb (MIC = 1–16 µg/mL), and a MDR MTb strain (MIC = 16 µg/mL)	[26]
2.8–8 nm	Spherical	Chemically synthesized	Activity against XDR MTb and MTb (strain H37Rv) (MIC = 1 μg/mL for both strains), and against a MDR-MTb strain (MIC = 4 μg/mL)	[19]
11–17.5 nm	Spherical	Chemically synthesized as nanocomposites of chitosan	Activity against MTb (strain H37Ra, MIC = 1.95 µg/mL)	[27]

MDR, multidrug resistant; MIC, minimum inhibitory concentration; MTb, *Mycobacterium tuberculosis*; NPs, nanoparticles; NR, not reported; NTM, Non-tuberculous mycobacteria; Ref., reference; XDR, extensively drug resistant.

**Table 3 antibiotics-09-00569-t003:** Parameters affecting mycogenic synthesis of MNPs [17,52,80].

Parameter	Effects	Consequences
pH	Formation of nucleation centers, time of reaction and morphology (size).	Alkalinization increases the number of nucleation centers, leading to enhanced formation of MNPs. Acidic pH (<2) reduces the formation of nucleation centers, leading to aggregation and bigger sized MNPs. At alkaline pH, protons and metal ions compete to establish bonds with negatively charged regions resulting in faster synthesis. Increasing pH decreases MNPs size.
Temperature	Resultant morphology (size and shape), synthesis rate, and formation of nucleation centers.	Higher temperatures lead to more uniform shapes, smaller sizes and better yields of most common MNPs. Increase in temperature, increases the rate of formation.
Time of reaction	Resultant morphology (size).	Increase in reaction time, increases the MNPs size.
Concentration of chemical precursors	Resultant morphology (size).	Smaller concentrations tend to lead to smaller MNPs and increased dispersion. Higher concentrations can generate increased toxicity.
Culture media	Quantity of MNPs.	Media containing enzymes-specific substrates increases enzyme production, which can generate more MNPs.
Quantity of fungal biomass	Quantity of MNPs.	More biomass leads to increased enzyme release, facilitating the mycosynthesis of MNPs and increasing their production.
Agitation	Resultant morphology (size), quantity of MNPs, and synthesis rate.	Little or no agitation decreases the synthesis rate and might lead to agglomeration, increased sizes, and reduced MNP production.
Light intensity	Quantity of MNPs and synthesis rate.	Light stimulates fungal growth and metabolite production. Higher metabolite concentration results in faster synthesis and increased MNP production.

**Table 4 antibiotics-09-00569-t004:** Mycogenic AgNPs with reported activity against mycobacteria.

AgNPs Size	AgNPs Shape	Fungal Species Involved in the Mycogenic Synthesis	Highlights	Ref.
3–20 nm	Spherical	*Rhizopus stolonifer* (filamentous fungus; family Mucoraceae)	Activity against clinical isolates of MDR MTb (MIC = 6.25–12.5 µg/mL)	[107]
22–50 nm	Spherical with agglomeration	*Trichoderma* sp. (filamentous fungus; family Hypocreaceae)	Activity at all concentrations tested (0.1, 0.5 and 1 ppm) against *M. smegmatis* (strain mc^2^155)*,* and *M. marinum,* being higher for *M. smegmatis*. Reduced the survival of intracellular (RAW264.7 macrophages) *M. smegmatis* (in 35%)*,* and *M. marinum* (in 5%)	[108]
≈17 nm sized silver chloride (AgCl) NPs	Spherical	Commercial yeast extract	AgClNPs (37 μg/mL), with activity against *M. smegmatis* (strain mc^2^155) and MTb (strain H37Rv)	[109]

MDR, multidrug resistant; MIC, minimum inhibitory concentration; MTb, *Mycobacterium tuberculosis*; NPs, nanoparticles; Ref., reference.

**Table 5 antibiotics-09-00569-t005:** MNPs (other than AgNPs) with reported activity against mycobacteria.

MNP Type	MNP Morphology	Synthesis Process	Highlights	Ref.
Gold (AuNPs)	15–30 nm sized, spherical and monodispersed	Chemically synthesized, stabilized with citrate, and Poly-allylamine hydrochloride (PAH)	Both citrate-AuNPs and PAH-AuNPs have activity against *M. bovis* (strain BCG), lower than tested AgNPs	[66]
Copper (CuNPs)	NR	Biologically synthesized from leaves of *Psidium guajava* L.	Activity against MTb, *M. smegmatis*, and *M. pheli*, but lower than other MNPs	[57]
Gallium (GaNPs)	305 nm sized and cylindrical	Chemically synthesized by double emulsification and sonication	Polydispersed, with prolonged activity against intracellular *M. smegmatis*	[113]
Copper oxide and zinc oxide (Cu(II)ONPs and ZnONPs)	Spherical	Chemically synthesized	Activity against *M. avium* subsp. *paratuberculosis*	[69]
Bimetallic Silver-Gold (Au-AgNPs)	10–70 nm sized and polydispersed	Phyto-synthesized from *Barleria prionitis*	*In vitro* activity against MTb (strain H37Ra, active–MIC = 0.06–0.12 μg/mL, and dormant–MIC = 1.05–2.53 μg/mL) and *M. bovis* (strain BCG, active–MIC = 0.32–0.42 μg/mL, and dormant–MIC = 0.32–0.64 μg/mL), as well as *ex vivo* in THP-1 cells infected with MTb (active–MIC = 0.63–1.46 μg/mL, and dormant–MIC = 0.56–2.16 μg/mL) Higher activity than AuNPs or AgNPs (MIC ≈ 2.5 μg/mL), and more specific for mycobacteria with a higher selectivity index In addition, the smaller MNPs (from *S. cumini*) are more effective	[60]
	90 nm sized and hexagonal	Phyto-synthesized from *Plumbago zeylanica*		
	10–20 nm sized and spherical	Phyto-synthesized from *Syzygium cumini*		
Zinc oxide (ZnONPs)	12–53 nm sized and spherical	Biologically synthesized from leaves of *Limonia acidissima* Linn. also known as *Feronia elephantum* Correa or wood apple	Activity against MTb (strain H37Rv)	[114]
GaNPs	≈300 nm sized and cylindrical	Chemically synthesized by double emulsification and sonication	Activity against intracellular MTb (strain H37Ra) in monocyte-derived macrophage (MDMs) and THP-1 macrophages (up to 70% MTb growth inhibition)	[115]
Zinc (ZnNPs)	≈60 nm sized and variable shapes, mostly spherical	Biologically synthesized from *Pseudomonas hibiscicola*	Activity against MTb (strain H37Rv) and a clinical MDR MTb strain, (MIC = 1.25 mg/mL for both strains)	[78]
Titanium dioxide (TiO_2_NPs)	16 nm sized and spherical	Chemically synthesized	Inhibited the growth of a clinical isolate of MTb (61%), and a clinical isolate of *M. bovis* (74%), at a concentration of 100 μg/mL. Effective surface coaters on inhibiting mycobacterial biofilm formation	[116]
Zinc oxide (ZnONPs)	5.4–13.2 nm sized and spherical	Chemically synthesized	Activity against MTb (strain H37Rv) and XDR MTb strains (MIC = 1 μg/mL for both), and MDR MTb (MIC = 4 μg/mL)	[19]
Selenium (SeNPs)	Spherical	Chemically synthesized	Activity against *M. smegmatis* (MIC = 0.4 μg/mL), and MTb (MIC = 0.195 μg/mL). Low toxicity (compared to other MNPs) and involvement in reduction of the integrity of the mycobacterial cell envelope. Colloidally stable	[117]

MDR, multidrug resistant; MIC, minimum inhibitory concentration; MNPs, metallic nanoparticles; MTb, *Mycobacterium tuberculosis*; NPs, nanoparticles; NR, not reported; Ref., reference; XDR, extensively drug resistant.

**Table 6 antibiotics-09-00569-t006:** Antimycobacterial activity complemented or increased by MNPs.

MNP Type	MNP Morphology	Combination	Highlights	Ref.
AgNPs	250–300 nm sized and spherical	With commercial Titanium dioxide (TiO_2_) NPs	Activity against *M. smegmatis* mc^2^155 (MIC > 100 μg/mL for 10:1 ratio, and MIC = 5 ± 2.4μg/mL for 50:1 ratio), and *M. bovis* (strain BCG, MIC = 11 μg/mL for 50:1 ratio)	[63]
AgNPs	5–50 nm sized	25 mg/Kg of AgNPs with 50 mg/Kg isoniazid	*In vivo* activity, in mice infected with a MDR MTb strain, led to a higher survival rate of 95%	[118]
AgNPs	22–50 nm sized, spherical with agglomeration	With cationic antimicrobial peptides, NK-2 (7 μg/mL) and LLKK-18 (1 μg/mL) at sub-lethal doses	Activity against *M. smegmatis* mc^2^155 and *M. marinum* increased after conjugation	[108]
		With rifampicin (RIF; 0.7 μg/mL)	Activity against *M. smegmatis* mc^2^155 and *M. marinum* increased after conjugation	
AgNPs	≈70 nm sized	With chloroform	Increased antimycobacterial activity when compared to AgNPs without chloroform	[41]
FeNPs nanocomposites	Magnetic	With chitosan and loaded with streptomycin	Successfully used against MTb and other microorganisms, showing higher activity than FeNPs and even FeNPs nanocomposites with chitosan	[119]
AuNPs	52.8 ± 5.33 nm sized and hexagonal	With both D- and L-enantiomeric forms of the amino acid serine	Active against *M. smegmatis* (strain mc^2^155) showing higher activity than D-serine alone, which is also known to be active against other mycobacterial species	[120]
AgNPs	30–80 nm sized	With ZnONPs	Increased MIC, when compared with the individual NPs, against MTb (strain H37Rv) both *in vitro* and *ex vivo* assays using THP-1 cells	[121]
AgNPs	17 ± 3 nm sized and spherical	Conjugated with vancomycin (VAN) (increased size of 30 ± 3 nm)	Activity against *M. smegmatis* with improved cell internalization by the conjugate in comparison with AgNPs and VAN on their own	[122]
AgNPs	20 nm sized and spherical	With ZnONPs	*Ex vivo* (in THP-1 macrophages) activity against MTb	[123]
		With ZnNPs and RIF	Increased (76% more than RIF on its own) *ex vivo* (in THP-1 macrophages) activity against MTb	
AuNPs	15 ±2 nm sized and spherical	With partial peptide tagged on the surface	Increased activity when compared to the peptide or the AuNPs on their own The attachment of the peptide increased the size of the NPs (to 20 ± 4 nm) and the inhibition of intracellular MTb (strain H37Rv) growth, from 45% for the peptide, to 59% for the AuNPs, to 91% for the peptide-AuNPs (all at 1 μg/mL)	[46]
AgNPs	2.8–8 nm sized and spherical	With ZnONPs (ratios 5:5, 2:8, 8:2, 3:7 and 7:3)	Activity against XDR MTb (MIC = 1 μg/mL for all ratios tested), MTb (strain H37Rv, MIC = 1–32 μg/mL), and against a MDR MTb strain (MIC = 4–64 μg/mL)	[19]

MDR, multi-drug resistant; MIC, minimum inhibitory concentration; MNPs, metallic nanoparticles; MTb, *Mycobacterium tuberculosis*; NPs, nanoparticles; NR, not reported; Ref., reference.

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
