# Peer review of "Mycogenic Metal Nanoparticles for the Treatment of Mycobacterioses"

_antibiotics, 2020, doi:10.3390/antibiotics9090569_

Round 1

Reviewer 1 Report

The manuscript “Mycogenic metal nanoparticles for the treatment of mycobacteriosis” reports a review on an interesting topic of mycogenic metal nanoparticle synthesis, and application for diseases caused by mycobacteria. The paper unfortunately is not well structured and leaves the reader lost in the unstructured/unnecessary details. Less is more, is a very appropriate characterization for the paper and how it should be improved. It seems that the authors were in rush to publish this review article. The article lacks updates reports on the particular topics. The most serious issue is structure of the article, which is not acceptable for an attractive review article.

Line 92-96, “Some of the clinical symptoms of TB are fever, cough, and dyspnea, amongst others. These are also the clinical symptoms observed in most patients with COVID-19 which can easily lead to misdiagnosis and ineffective treatments. Similar symptoms constitute a major challenge when controlling both Tb and COVID-19.” This statement is misleading, and if true, needs strong support of evidences.

Line 343-346, statement needs proper citation.

The section ‘other MNPs against mycobacteria’ needs more focus.

There are several grammatical and typo errors.

Author Response

1) The manuscript “Mycogenic metal nanoparticles for the treatment of mycobacteriosis” reports a review on an interesting topic of mycogenic metal nanoparticle synthesis, and application for diseases caused by mycobacteria. The paper unfortunately is not well structured and leaves the reader lost in the unstructured/unnecessary details. Less is more, is a very appropriate characterization for the paper and how it should be improved. It seems that the authors were in rush to publish this review article. The article lacks updates reports on the particular topics. The most serious issue is structure of the article, which is not acceptable for an attractive review article.

The structure of the manuscript was revised and changed according to all reviewers’ comments. The manuscript was carefully reviewed and complemented wherever found necessary.

2) Line 92-96, “Some of the clinical symptoms of TB are fever, cough, and dyspnea, amongst others. These are also the clinical symptoms observed in most patients with COVID-19 which can easily lead to misdiagnosis and ineffective treatments. Similar symptoms constitute a major challenge when controlling both Tb and COVID-19.” This statement is misleading, and if true, needs strong support of evidences.

This statement is now supported with more data and references. Now in lines 119-130.

3) Line 343-346, statement needs proper citation.

The statement is now cited with the relevant and appropriate references. Now in lines 315-322.

4) The section ‘other MNPs against mycobacteria’ needs more focus.

We have revised this section. Now in lines 331-339.

5) There are several grammatical and typo errors.

The manuscript was carefully revised, and all detected typos and errors were corrected.

Reviewer 2 Report

This is a review article, covering some of the literature of the last five/ten years, on the scientific research carried out in relation of metal nanoparticles (especially silver nanoparticles) against mycobacteriosis. The presentation is good, the review structure is correct and "Antibiotics" is an adequate journal for its publication. This reviewer recommends its publication as it is.

Author Response

The authors thank the reviewer for understanding the goal of this review and for considering it relevant and appropriate.

Reviewer 3 Report

The aim of the review is to present the extensive research and application of mycogenic metal nanoparticles, with a focus on silver nanoparticles, synthesized by filamentous fungi, especially against mycobacterial infections. It is a subject with great interest, especially nowadays when a pandemic disease is spread all over the world and the interest in finding a cure is tremendous.

In order to publish this paper in the journal, major corrections are needed and my recommendations are:

  1. Being a review paper, please indicate the methodology of finding articles, indicate the data bases used, the number of studied articles, the reviewed period, etc
  2. Introduction chapter must be reorganized; introduction as such must be expanded
  3. All the sub chapters belonging to chapter 1 must be renumbered; they seem to be belonging also for chapter 1 which is an introductory chapter
  4. The title conduct the readers to the idea that the article is about “Mycogenic metal nanoparticles for the treatment of  mycobacteriosis” and the chapter dedicated to this subject is very poor described (only 9 articles used as  direct references in 1.4 and 4 articles used as  direct references in 1.5); only this sub chapters 1.4 and 1.5 represent the core of the article with direct relation with the title; in sub chapter 1.5.1 only 8 references for AgNPs against MTb and in sub chapter 1.5.2 only 1 reference for Mycogenic AgNPs against MTb
  5. A table to summarize the described info from actual chapter 1 is needed (with info such as type of NP, form and dimensions; obtaining method for NP; the effects of NP, etc)
  6. All over the manuscript there is a lack of personal opinion of the authors regarding the described cases.
  7. Overall, the number of references must be extended; for a such generous topic only 61 references are extremely few.
  8. The conclusion chapter must be extended

Author Response

The aim of the review is to present the extensive research and application of mycogenic metal nanoparticles, with a focus on silver nanoparticles, synthesized by filamentous fungi, especially against mycobacterial infections. It is a subject with great interest, especially nowadays when a pandemic disease is spread all over the world and the interest in finding a cure is tremendous. In order to publish this paper in the journal, major corrections are needed and my recommendations are:

  1. Being a review paper, please indicate the methodology of finding articles, indicate the data bases used, the number of studied articles, the reviewed period, etc.

This information was now included in the manuscript as a methods section.

  1. Introduction chapter must be reorganized; introduction as such must be expanded

The entire manuscript was revised, and its structure changed.

  1. All the sub chapters belonging to chapter 1 must be renumbered; they seem to be belonging also for chapter 1 which is an introductory chapter

The entire manuscript was revised, and its structure changed as well as the numbering of the sections.

  1. The title conduct the readers to the idea that the article is about “Mycogenic metal nanoparticles for the treatment of mycobacteriosis” and the chapter dedicated to this subject is very poor described (only 9 articles used as direct references in 1.4 and 4 articles used as direct references in 1.5); only this sub chapters 1.4 and 1.5 represent the core of the article with direct relation with the title;

There is little research reported on this specific topic. This is the reason that led the authors to focus on it, since they consider this to be a relevant and underexplored potential solution for the control of general mycobacteriosis and tuberculosis. Nevertheless, a more exhaustive list of references was used for this version of the manuscript.

  1. in sub chapter 1.5.1 only 8 references for AgNPs against MTb and in sub chapter 1.5.2 only 1 reference for Mycogenic AgNPs against MTb

There is extensive research on MNPs and on mycogenic MNPs, but, as mentioned before, little has been investigated regarding mycogenic MNPs against mycobacteria. All reports found were mentioned in the manuscript.

  1. A table to summarize the described info from actual chapter 1 is needed (with info such as type of NP, form and dimensions; obtaining method for NP; the effects of NP, etc)

The authors have followed the suggestion made and are now presenting part of the data in tables.

  1. All over the manuscript there is a lack of personal opinion of the authors regarding the described cases.

The discussion section was extended, and the authors point of view is now presented in a more clear and assertive way.

  1. Overall, the number of references must be extended; for a such generous topic only 61 references are extremely few.

The authors have complemented the manuscript with more details whenever found necessary, and more references were included.

  1. The conclusion chapter must be extended

The conclusion section was extended.

Round 2

Reviewer 1 Report

This revised version of review on “Mycogenic metal nanoparticles for the treatment of mycobacteriosis” is not seriously revised. The manuscript is written like a research article divided in sections like methods, results and discussion. This is misleading and inappropriate style for a review article.  Again, I would comment that "it seems that the authors are in rush to publish this review article" which is not good for science. I would suggest to the authors to please read some good review articles and follow the style; method section must be removed, result and discussion (without data, are you serious?!). I appreciate the effort for updating recent work and wish could put more effort on structure of the article.  

Author Response

Reviewer 1:

This revised version of review on “Mycogenic metal nanoparticles for the treatment of mycobacteriosis” is not seriously revised. The manuscript is written like a research article divided in sections like methods, results and discussion. This is misleading and inappropriate style for a review article. Again, I would comment that "it seems that the authors are in rush to publish this review article" which is not good for science. I would suggest to the authors to please read some good review articles and follow the style; method section must be removed, result and discussion (without data, are you serious?!). I appreciate the effort for updating recent work and wish could put more effort on structure of the article.

Following the first round of revision, our manuscript is now in accordance with PRISMA, transparent reporting of systematic reviews and meta-analyse checklist, which justifies some of the issues raised regarding the structure of the paper. The adoption of PRISMA is the reason why the review includes a methods section, as well as one for results and another for discussion. The data being discussed comprehends all the information from the published papers being analyzed.

Nevertheless, we appreciate all comments that can help us improve our manuscript.

In the current revision we have further streamlined the manuscript and have improved its structure and readability.

Reviewer 3 Report

The authors extensively modified the manuscript, answering the reviewers' comments. I consider that the manuscript can be accepted for publication, after a minor revision, considering the following aspects:

1.      In the chapter “methods” it is welcomed to insert for each key term the number of articles consulted from databases

2.       At chapter 3.2.1. B) AgNPs against MTb please insert in the text a reference to table 4.

3.      At chapter 3.2.3. MNPs (other than AgNPs) against mycobacteria please insert in the text a reference to table 6.

4.      In the manuscript , at first appearance, please use the complete name of the bacteria, fungi, etc, and after that, use only the abbreviation. 

Author Response

Reviewer 3:

The authors extensively modified the manuscript, answering the reviewers' comments. I consider that the manuscript can be accepted for publication, after a minor revision, considering the following aspects:

  1. In the chapter “methods” it is welcomed to insert for each key term the number of articles consulted from databases

The adoption of PRISMA was only considered after the first round of revision, which prevents us from accurately providing exact numbers, so we have opted for not including this level of detail in our revision.

  1. At chapter 3.2.1. B) AgNPs against MTb please insert in the text a reference to table 4.

We have now included a reference to this table (now table 2) in the text (line 254).

  1. At chapter 3.2.3. MNPs (other than AgNPs) against mycobacteria please insert in the text a reference to table 6.

We have now included a reference to this table (now table 5) in the text (line 345).

  1. In the manuscript, at first appearance, please use the complete name of the bacteria, fungi, etc, and after that, use only the abbreviation.

We have revised the scientific names throughout the manuscript.